# BIPS—A code base for designing and coding of a Phage ImmunoPrecipitation Oligo Library

**Sigal Leviatan**[1,2☯], **Iris N. Kalka**[1,2☯], **Thomas Vogl**[1,2], **Shelley Klompas**[1,2], **Adina Weinberger**[1,2], **Eran Segal**[1,2]*

**1** Department of Molecular Cell Biology, Weizmann Institute of Science, Rehovot, Israel, **2** Department of Computer Science and Applied Mathematics, Weizmann Institute of Science, Rehovot, Israel

☯ These authors contributed equally to this work.
* eran.segal@weizmann.ac.il

**Data Availability Statement:** All relevant data are within the manuscript and its Supporting Information files.

## Abstract

BIPS (Build Phage ImmunoPrecipitation Sequencing library) is a software that converts a list of proteins into a custom DNA oligonucleotide library for the PhIP-Seq system. The tool creates constant-length oligonucleotides with internal barcodes, while maintaining the original length of the peptide. This allows using large libraries, of hundreds of thousands of oligonucleotides, while saving on the costs of sequencing and maintaining the accuracy of oligonucleotide reads identification. BIPS is available under GNU public license from: https://github.com/kalkairis/BuildPhIPSeqLibrary.

This is a *PLOS Computational Biology* Software paper.

## Introduction

The use of Phage immunoprecipitation sequencing (PhIP-Seq), based on phage-displayed synthetic oligonucleotide libraries [1], became a useful tool in scanning the reaction of the human immune system to large protein collections [2–5]. PhIP-Seq allows testing for antibody reaction to hundreds of thousands of antigens in parallel, in contrast to peptide arrays and enzyme-linked immunosorbent assays (ELISAs) which have limited capacities, of at most a few thousand or a few tens of thousands of antigens [6]. However, creating a library of that many peptides is computationally challenging.

Current state-of-the-art software for peptide library design includes Pepsyn [7], which uses representative selection for tiling protein sequences. Other previous works [2,8] did not publish public software, but mention crucial steps in constructing peptide libraries. While Pepsyn is a reliable tool, it does not allow for representations of multiple variants of the same peptide sequence.

Here, we present BIPS (BuildPhIPSeqLibrary), a software for rapid and simple construction of an oligonucleotide library from any input sets of proteins (Fig 1A). BIPS is intended for large peptide libraries and through the use of synonymous mutations, which was previously suggested [2,8], enables the complete representation of multiple variants in an unambiguous manner, while allowing peptide identification without the need for full-length sequencing.

**Funding:** The author(s) received no specific funding for this work.

**Competing interests:** The authors have declared that no competing interests exist.

We tested BIPS on simulated data, on all available infectious and allergic diseases from the Immune Epitope Database (IEDB). Through these, we showed the computational limits of BIPS.

**Fig 1. Library construction and barcoding schematic representation. a,** Flow of inputs from peptide sequences, through software modules (arrows) including intermediate results. (1) protein sequences splitting into constant-length peptides with overlap. (2) reverse translation adhering to codon usage frequencies. (3) barcoding sequences' tails. **b,** Three possible identification methodologies of oligos. In brown below are minimal sequencings required for correct identification. (1) External barcodes decrease the possible lengths of coding peptides via replacing the 3' end with a short barcode. (2) No barcoding, requires sequencing of entire oligonucleotides. (3) Inline barcoding, requires intermediate length sequencing, and enables full-length coding peptides. **c,** Two possible inline barcodes of the same 5' sequence (LL), with a hamming-distance of three and a six-nucleotide barcode (3x2 nucleotides). First oligonucleotide is barcoded without restrictions. Second oligonucleotide is restricted for three iterations, and permitted in the fourth. Figures were created with BioRender.com.

## Design and implementation

### Design

An overview of BIPS is presented in Fig 1A, and is detailed below.

Construction of PhIP-Seq libraries begins with a large set of proteins provided by users. Usually these will be full-length or partial proteins, from one or many sources. Note that the advantages of a PhIP-Seq library are more pronounced when exact epitopes are unknown and a scanning of relatively long peptides or full proteins is required.

In constructing a PhIP-Seq library the set of proteins must be cut into constant length peptides, so as not to create biases in PCR-amplification of the library and its cloning into phages. To this end, shorter input proteins or peptides are terminated by a stop codon, and padded with a random amino-acid sequence, as implemented in Pepsyn [7].

The PhIP-Seq library is intended to present all potential linear epitopes (potential binding sites of the immune system antibodies) of the given set of proteins. In order to cover all potential linear-epitopes with potential for antibody binding, proteins are cut with a positive overlap. This overlap should be long enough so that linear epitopes, usually 6–20 amino acids long [9] appear in full in at least one of the peptides.

BIPS produces a list of oligonucleotide sequences (oligos), to be synthesized and cloned into phages, that will infect and replicate in the appropriate bacteria. The common use is of bacteriophage T7 within *Escherichia coli*. All peptides must be coded using the host bacteria's codon usage table, without including restricted sequences. Furthermore, in order to make the replication as efficient and as uniform as possible, it is preferable to also imitate the codon usage frequencies of the host bacteria (Fig 2A).

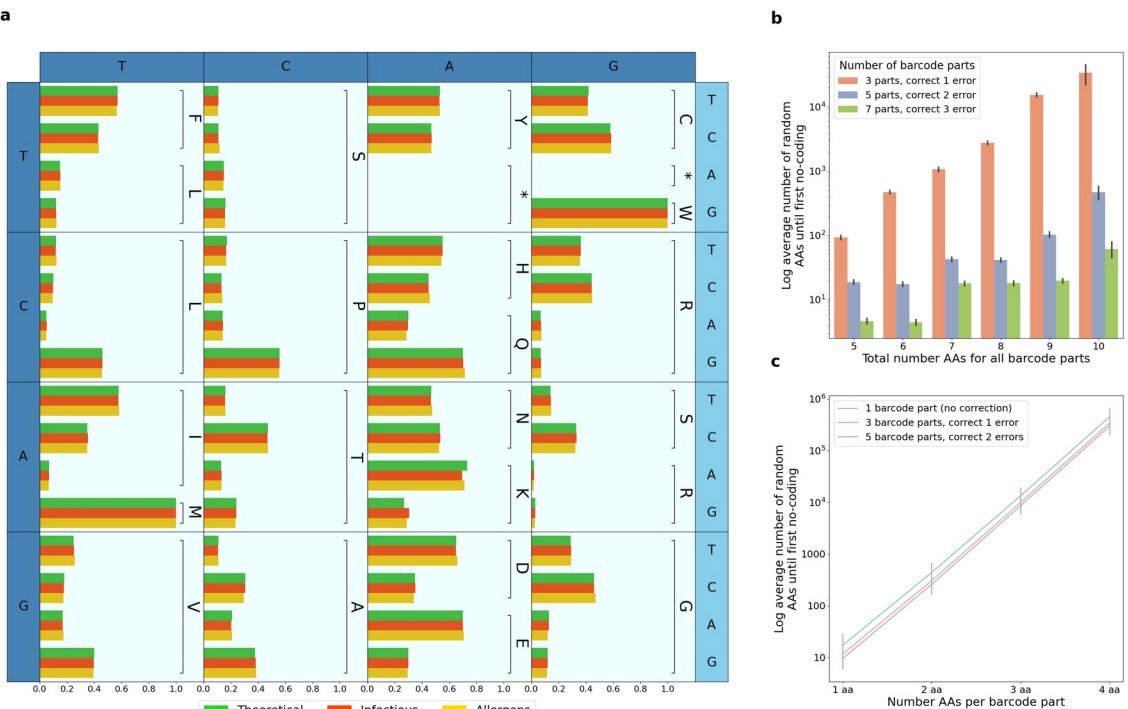

**Fig 2. Codon frequency in BIPS output. a**, Codon frequencies (summing to one per amino acid). In green the published frequencies for Escherichia coli. In red and yellow are coding frequencies of example outputs from IEDB's infectious diseases and allergic diseases. **b**, **c**, Simulations of average number of barcodes for random oligos. Colors are defined by barcode-lengths and number of barcode parts. Simulations were performed on constant barcode length (**b**) or constant barcode parts' lengths (**c**).

After immunoprecipitation, the peptides in the bound phages are identified by sequencing the bound phages. In cases of libraries where there are many similar peptides due to research of deep mutagenesis [3], mutation scanning, homologous proteins or other, partially sequencing a peptide might not be sufficient for identification, and alternative methods must be explored. One method of identification is sequencing of entire oligos, however this is costly and prone to read errors. Another method is attachment of pre-planned barcodes at the 3' end of each oligo, preceded by a stop codon, so that the barcode does not translate into a protein. Such barcodes are relatively short in length, thus requiring cheaper sequencing, and are designed to correct read errors via their pre-planned hamming-distance [10] from each other. These advantages come at a cost, the length of the barcode is added upon the length of the coding oligo, adding to the costs of library synthesis. Worse, this addition limits the length of actual coding oligo, as DNA synthesis technology is still limited. BIPS creates internal barcoding sequences by leveraging the multiple possibilities in coding amino-acids into oligos, while maintaining a requested hamming-distance between pairs. Barcodes are created at the coding stage as part of the oligo, thus they can be located at the 5' end of the oligo eliminating the mis-identification problem caused in cases of multiple inserts (S2 Text).

If a constructed barcode does not adhere to the hamming-distance or the entire peptide includes restricted sequences, BIPS re-codes the barcode until either getting a successful barcode, or until reaching a pre-defined number of iterations. The concept of utilizing synonymous codons to allow unambiguous mapping has previously been discussed [2,8], here, we have advanced this approach by using it in order to ensure a hamming distance between sequences. It is clear that coding many identical (or nearly identical) sequences in a single library will increase the use of relatively rare codons, however we have shown [4] that this does not cause a significant reduction in differential expression.

When oligo libraries are large (hundreds of thousands of oligos) it is computationally infeasible to check the hamming distance of each newly proposed oligo vs. the library already created, therefore we construct a unique divide-and-conquer method to ensure the hamming distance. Given a required number of errors to correct n (usually one or two errors), and the respective hamming distance of $2n+1$ (three or five respectively), we split the barcode into $2n+1$ parts, and demand uniqueness for each of the parts. Uniqueness is less computationally complex to check, and ensures at least one difference per part, thus at least $2n+1$ differences overall (i.e., a hamming distance of at least $2n+1$). Such internal barcodes are longer than pre-planned barcodes, however they allow using the full length of the oligo sequence as coded peptide. While in comparison to not barcoding at all, this methodology removes the need to sequence the entire oligo in order to identify it (Fig 1B and 1C).

At the final stage of preparation, a constant prefix and suffix are added to each finalized oligo sequence, these include the restriction sites and primer binding region for library amplification. In order to produce the protein, a stop codon must appear downstream of the peptide, it is suggested to include this within the suffix, (at 3' end), but it is not mandatory (in case C-terminal tags should be added or M13 phage display should be used). It is important to define in advance all relevant restriction sequences (including these restriction sites), so as not to allow them to be formed when coding the peptides into oligo, including in the barcodes.

We present BIPS, a configurable software to build PhIP-Seq libraries. It receives sets of proteins (or partial proteins) as inputs, creating a library of internally barcoded oligos, spanning the entire input protein set. Final oligos are marked with all the original proteins and locations they were derived from, and mapped to all proteins containing them (S3 Text), allowing for the analysis of the bound peptides and their source proteins. There is still a lot of room for exploration via PhIP-Seq of antibody repertoires, and BIPS facilitates achieving this goal.

## Software implementation

All code was written in Python 3, and tested on multiple platforms (CentOS, IOS and Windows all in the Anaconda environment). The code consists of two main programs, the first converts multiple files of proteins to a single file of oligos for synthesis, and the second maps oligos to original files and peptides. The latter program takes a relatively long time, however it can be run offline. Both programs can utilize multithreading to reduce run-time, if available.

All functions in the code have accompanying unit-tests. To ensure all dependencies and code versions are in order, it is advisory to run all tests on the code on the available platform.

The program converting protein files to oligos takes as input from the configuration the path to a directory, where all protein files should be located. It is possible to add new protein files and re-run the program, adding their accompanying oligos to the library. However, it is not possible to change an existing input file without erasing the entire output directory first, and re-starting the library build. This limitation exists to maintain consistency between the input files and the complete library. At the end of the run a configured output directory will contain all output files created during the run. The different output files are described in the README.md file of the Data/Output directory, the most important of which is order_file.txt, a ready to order file containing only nucleotide sequences.

The mapping program runs on two files: oligos_sequence.csv and sequences_ids.csv, both of which are created in the library constructing program. The program goes over all oligos and all protein sequences and maps the oligos to all possible perfect matches. This program produces a single mapped oligo file in the output directory under the name mapped_oligos_sequence.csv.

After producing the library and performing PhageIPSeq experiments FASTQ files should be returned from sequencing machines. We supply an example code to identify the barcode. We note that this specific code example is configured to default configurations and should provide a roadmap to identifying sequencing results rather than be used directly.

## Barcode length and error correction

Our unique divide-and-conquer method allows a low-computational method of ensuring the required hamming distance between every pair of barcoded oligos.

The idea is to ensure the correction of up to n-errors by using barcodes at hamming distance ($2n+1$) from each other, by dividing the length of the barcode in ($2n+1$) parts, as equally as possible, and never having two oligos with an identical sub-sequence at any of the barcode parts. Since checking for identity is easy (using a hash function, or inherently in code structures like python set) then it is computationally easy to ensure each part has at least one difference from any other oligo, and together at least ($2n+1$) differences.

For example, a barcode length of 75 bps, divided into 5 sections of correcting for 2 read errors, and a random protein library, millions of oligos can be barcoded in this way (at most $4^{15} \approx 10^9$, i.e. a billion). In practice, due to the uneven amino acid probabilities, and the amount of redundancy in the specific library, it is possible that some sequences will fail to code, for example there is only one way to code "MMMMM" so if two peptides beginning with "MMMMM" exist in the library and the first part of the barcode is of length 15 base pairs (5 amino acids), then the second peptide will not be codable.

In essence there are three parameters here that need to be balanced, the length of the barcode (and thus the length one needs to sequence after immuno-precipitation), the number of errors one wants to correct, and the size and richness of the library. If the library contents are not diverse (for example a database which includes many versions of the same protein, e.g.

many wheat protein variants as may appear in an allergens database), then a longer barcode will be needed to allow enough coding options.

## Configuration

A single configuration file allows defining all parameters: length of peptide, and overlap, in amino acids, location and lengths of barcode parts, codon table and frequencies, restricted sequences, and a prefix and suffix for all final oligonucleotides. As well as the input and output directories.

## Results

We tested the performance of BIPS on both simulated and real data.

### Barcode construction simulations

We simulated library and barcode construction to evaluate upper bounds of possible barcodes in a library. To that end, we first created random amino-acid sequences by sequentially randomly choosing each amino acid from E. coli amino-acid frequencies. From those we constructed barcodes for each sequence. Upon the first case of barcode construction failure (meaning that at least one of the barcode parts was non-unique) we halted the simulations. If there were no barcode construction failures, we halted each single simulation upon reaching $10^7$ sequences.

We note that all simulations, although providing insights about barcoding limitations, do not represent real life protein sequences. That is, in common proteins, the amino-acid order is non-random, and furthermore they often include recurring motifs.

For the first set of simulations (Fig 2B), for each total barcode length we created barcodes split into 3, 5 and 7 parts. These in turn allow correction of 1, 2 or 3 errors respectively. We performed the simulations for total barcode lengths of 5aa-10aa (15–30 base-pairs). The split of barcode of total length $l$ base pairs into $m = 2n+1$ barcode parts was performed by use of $\frac{l}{m} + \text{II}(i < (l\%m))$ base pairs for barcode part $i$. We note that some of the splits do not necessarily start or end at an amino-acid boundary (e.g., for a 6aa barcode of 5 parts barcodes will be [4, 4, 4, 3, 3] base pairs).

For the second set of simulations (Fig 2C), each simulation maintained a constant length of each barcode part. We performed the simulations for barcode part lengths ranging from 1aa to 4 aa. For each such simulation we constructed barcodes made of 1, 3 or 5 such parts. Note, that in this case, unlike in the first set of simulations, each barcode part includes complete amino-acid sequences and does not contain partial codons. Furthermore, in this simulation the barcode's total length is dependent on the number of barcode parts and is not constant.

### Performance on real data

For understanding BIPS performance on real data we obtained two separate datasets from IEDB, infectious diseases and allergic diseases (S1 Text). We ran BIPS separately on each of these inputs, creating two separate libraries (with default parameters including oligo_aa_-length = 64, oligo_aa_overlap = 20, and a 5-part barcode at the 5' end of 15bp per part). Codon frequencies were calculated on the oligo sequences without the amplification prefix and suffix. The frequencies were compared to theoretical frequencies [11] (Fig 2A).

For IEDB's infectious diseases: of the 1,249 full protein sequences two could not be used (uniprot A0A2D4C4Z9 contained a non-amino acid letter "X", uniprot Q8I4R2 had no amino acid sequence). The remaining 1,247 proteins were cut into a total of 24,456 peptides.

Of the 24,456 peptides, 51 failed in conversion to a nucleotide sequence, because of restriction sites. After the initial coding 400 had to be recoded because of barcode collisions, of them 180 failed to recode. This run took 7 minutes on CentOS.

For the IEDB's allergic diseases: Of the 240 proteins two could not be used (no amino acid sequence for uniprot M0U687 and Q9URH1). The remaining 238 proteins were cut into 1,745 peptides.

Of the 1,745 peptides, 1 failed in conversion to a nucleotide sequence, because of restriction sites. After the initial coding 10 had to be re-coded because of barcode collisions, all succeeded. This run took 10 seconds on CentOS.

## Availability and future directions

BIPS is open-source and freely available under the GNU public library; it is maintained on GitHub, enabling bug-reporting and community collaboration. It can be found at https://github.com/kalkairis/BuildPhIPSeqLibrary. The entire software was developed in Python 3, and includes internal tests and use-cases for the benefit of users.

We plan to use BIPS in order to design a new PhIP-Seq library for a new cohort within our group. In the future we intend to develop a publicly available software to analyze results from PhIP-Seq sequencings.

## Dependencies

Code uses standard Python 3 libraries, in addition to: pandas (1.3.5), numpy (1.21.5), regex (2022.1.18). Code has been tested on Windows, IOS and CentOS.

## Supporting information

**S1 Text. Detailed information of test data use for checking this software library.**
(DOCX)

**S2 Text. Explanation of the misidentification problems, which might occur when adding a barcode at the 3' end.**
(DOCX)

**S3 Text. Description of extra software supplied in order to identify the oligo source of each barcode sequenced, and all potential protein sources it may have been derived from.**
(DOCX)

## Author Contributions

**Conceptualization:** Sigal Leviatan, Adina Weinberger, Eran Segal.

**Data curation:** Thomas Vogl, Shelley Klompas, Adina Weinberger.

**Formal analysis:** Sigal Leviatan, Iris N. Kalka.

**Methodology:** Sigal Leviatan, Iris N. Kalka.

**Software:** Sigal Leviatan, Iris N. Kalka.

**Supervision:** Eran Segal.

**Visualization:** Sigal Leviatan, Iris N. Kalka.

**Writing – original draft:** Sigal Leviatan, Iris N. Kalka.

**Writing – review & editing:** Thomas Vogl, Eran Segal.

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
