## [Decision Letter · Decision Letter 0]

3 Jul 2022

Dear Ms Leviatan,

Thank you very much for submitting your manuscript "BIPS - a Code Base for Designing and Coding of a Phage Immuno-Precipitation Oligo Library" for consideration at PLOS Computational Biology. As with all papers reviewed by the journal, your manuscript was reviewed by members of the editorial board and by several independent reviewers. The reviewers appreciated the attention to an important topic. Based on the reviews, we are likely to accept this manuscript for publication, providing that you modify the manuscript according to the review recommendations.

Sincerely,

Dina Schneidman

Software Editor

PLOS Computational Biology

[LINK]

Reviewer's Responses to Questions

**Comments to the Authors:**

Reviewer #1: BIPS is a comprehensive tool to generate a Phage Immuno-precipitation library from Protein/peptide sequences of interest. The tool generates a library with desired oligo length and overlap.

Novelty of the code lies in the embedded barcode. In commonly used methods the barcode is at the 5p end of the sequences that reduces the effective length of the oligo or on the 3p end that makes paired end sequencing necessary.

This tool also ensures appropriate hamming distance between oligos.

This tool generates a barcode of desired length within the sequence with an elegant use of codon optimization table. Internal barcoding reduces sequencing length requirement.

I have some minor comments:

1. Does the codon optimization reduce the probability of expression of that oligo, if there are large number of similar sequences?

2. Would it be possible to use the code without using internal barcoding strategy?

Reviewer #2: Leviatan et al have developed an informatics pipeline for converting protein sequences into DNA sequences encoding peptides to tile across the desired proteins (BIPS). The DNA sequences are then to be cloned en masse into phage display for the PhIP-Seq assay. PhIP-Seq has emerged as a powerful antibody profiling technology, so open source tools for library design will certainly be welcomed by users – especially since there remains a lack of established computational tools and standards. However, there are several unfortunate shortcomings of this manuscript that must be addressed before it is suitable for publication.

Major Issues

• Perhaps the most critical problem with this paper is that the open-source pipeline currently in use for PhIP-Seq library design, pepsyn (Mohan et al. Nat Prot), is not even referenced. Since it is not referenced, BIPS cannot be directly compared to the de facto standard.

• A second critical issue is that codon-based barcoding has been incorporated into the design of multiple previously published PhIP-Seq libraries. It has been utilized in at least two contexts of which I am aware. In the first, Xu et al (Science) used codon barcoding to distinguish alanine scanning variants of public epitopes. In the second example, Shrock et al (Science) used codon barcoding to distinguish duplicate peptides that enhance assay robustness. Xu is referenced but not in the context of codon barcoding, and Shrock is not referenced at all. Since these prior examples are not referenced, the approach used by BIPS cannot be compared or contrasted.

• The overall rationale for embedding codon barcodes is not articulated. It seemed to me that the main reason the authors would use codon-barcoding is for disambiguation of similar library members. However, for typical libraries and typical read lengths (say 50 nucleotides at minimum, but typically longer reads at incremental additional cost), the number of ambiguous library members would be extremely small. Of course, any number of ambiguous sequences is undesirable, but rescuing these sequences might not be the most compelling rationale. I suggest the authors emphasize additional reasons why codon barcoding is useful. Mutation scanning, deep mutagenesis, and peptide duplication are what I think of as the most important examples, but there are likely others. Reducing read length requirements might also be compelling to some users, but more specifics should be provided on the actual benefit for standard libraries and standard read lengths.

• The authors are inconsistent with the PhIP-Seq abbreviation. PhageIP-Seq is used occasionally, but it is unclear if this is intentional or in error. The most commonly used long form seems to be Phage ImmunoPrecipitation Sequencing (with no hyphen like what is in the title).

Minor Issues

• In the first paragraph, it is mentioned that peptide arrays have at most a few thousand antigens. There are certainly examples of more than a few thousand antigen arrays.

• Padding shorter peptides with random amino-acid sequences downstream of a stop codon is also a feature highlighted in the pepsyn software so should be referenced as such.

• Typo “DNWA”

• It is mentioned that the 3’ suffix must start with a stop codon. However, many users may want to have an epitope tag or some other fusion peptide C-terminal to the peptide. Requiring a stop codon would also make the design incompatible with M13 phage display, which may also be of interest to some of your readers.

• It is not clear how restriction sites are avoided during the peptide coding process. Could you provide additional information?

• I had a hard time evaluating Fig 2b/c since the font was so small.

• I believe the Y-axis title of Fig 2b has a typo.

• The term “infectious diseases” is used where I think it is meant “Infectious organisms”.

• In the paragraph above “Code Availability” section, FASTA files are referred to, but I believe it is meant “FASTQ”.

• Define “IRIs”.

• At the end of the Test Case section, the number 24,456 appears to have been incorrectly copied from above. The replacement should be 1,745 if I am not mistaken.

• In the “Finding the Source of a Barcode” section, the authors suggest doing sequencing read mapping by alignment with errors allowed. However, certain oligo synthesis errors may prematurely truncate the peptide or be otherwise deleterious. So the pros and cons of taking this approach should be presented.

• Fig. 2 only referred to only in Supplementary/Methods?

• A typical peptide length and corresponding barcode length should be given with numbers. The qualitative descriptions are nice to read but some numbers to get an order of magnitude would be helpful.

• The sentence “Figures were created with Biorender.com” seems to be in the wrong place

• It seems in the text that the software has three options for barcoding, but after checking the code, it seems only be able to do the third. Please clarify.

• The example and default configurations should be mentioned in the methods text

• hamming distance (2n+1) … please explain n for the unfamiliar reader.

• Please clarify the meaning of this sentence: Due to computational limitations we performed each single simulation up to 107 sequences

• “Multiple inserts” should be defined for the unfamiliar reader.

**Have the authors made all data and (if applicable) computational code underlying the findings in their manuscript fully available?**

Reviewer #1: Yes

Reviewer #2: **No: **I did not attempt to confirm

PLOS authors have the option to publish the peer review history of their article (what does this mean?). If published, this will include your full peer review and any attached files.

Reviewer #1: No

Reviewer #2: No

Figure Files:

Data Requirements:

Reproducibility:

References:

---

## [Decision Letter · Decision Letter 1]

18 Sep 2022

Dear Ms Leviatan,

We are pleased to inform you that your manuscript 'BIPS - a Code Base for Designing and Coding of a Phage ImmunoPrecipitation Oligo Library' has been provisionally accepted for publication in PLOS Computational Biology.

Please address the Reviewer 2 comment in the final version.

Best regards,

Dina Schneidman

Software Editor

PLOS Computational Biology

Reviewer's Responses to Questions

**Comments to the Authors:**

Reviewer #1: The authors have comprehensively addressed my concerns. I believe they have also addressed those of the other reviewer. I commend them for this effort.

Reviewer #2: The authors have addressed the issues I raised. However, in testing their code, I discovered a flaw that either should be fixed. The peptide coding nucleotide sequences are selected to avoid certain sequences such as those that will be used for downstream cloning. Currently, this happens before addition of prefix and suffix sequences. It is therefore possible (and quite likely in fact) that the avoided sequences are inadvertently introduced at the junction of the prefix and peptide encoding sequence and/or at the junction between the peptide coding sequence and the suffix. This can of course lead to truncation or complete loss of peptides during the library cloning process.

Minor point. In response to one my comments, the authors write:

"We stand corrected. ELISAs are usually limited to a few hundreds or thousands of oligos, but other

peptide arrays can indeed get to tens of thousands."

I'm not sure what this is supposed to mean about ELISAs, since they don't use oligos and they typically are not done for more than a very small number of antigens (1 to 10). My point in the original critique was about peptide arrays. The authors said these are limited to tens of thousands, but I have seen some with many more (e.g. those produced with micromirrors and photochemistry). But it is true that most peptide or protein arrays are in the 10s of thousands of antigens or less.

**Have the authors made all data and (if applicable) computational code underlying the findings in their manuscript fully available?**

Reviewer #1: None

Reviewer #2: Yes

PLOS authors have the option to publish the peer review history of their article (what does this mean?). If published, this will include your full peer review and any attached files.

Reviewer #1: No

Reviewer #2: No

---

## [Editor Report · Acceptance letter]

3 Nov 2022

PCOMPBIOL-D-22-00602R1 

BIPS - a Code Base for Designing and Coding of a Phage ImmunoPrecipitation Oligo Library

Dear Dr Leviatan,

I am pleased to inform you that your manuscript has been formally accepted for publication in PLOS Computational Biology. Your manuscript is now with our production department and you will be notified of the publication date in due course.

With kind regards,

Zsofia Freund
